# Are Pre-trained Language Models Aware of Phrases? Simple but Strong Baselines for Grammar Induction

**Taeuk Kim[1], Jihun Choi[1], Daniel Edmiston[2] & Sang-goo Lee[1]**
[1]Dept. of Computer Science and Engineering, Seoul National University, Seoul, Korea
[2]Dept. of Linguistics, University of Chicago, Chicago, IL, USA
`{taeuk,jhchoi,sglee}@europa.snu.ac.kr, danedmiston@uchicago.edu`

## Abstract

With the recent success and popularity of pre-trained language models (LMs) in natural language processing, there has been a rise in efforts to understand their inner workings. In line with such interest, we propose a novel method that assists us in investigating the extent to which pre-trained LMs capture the syntactic notion of constituency. Our method provides an effective way of extracting constituency trees from the pre-trained LMs without training. In addition, we report intriguing findings in the induced trees, including the fact that some pre-trained LMs outperform other approaches in correctly demarcating adverb phrases in sentences.

## 1 Introduction

Grammar induction, which is closely related to unsupervised parsing and latent tree learning, allows one to associate syntactic trees, i.e., constituency and dependency trees, with sentences. As grammar induction essentially assumes no supervision from gold-standard syntactic trees, the existing approaches for this task mainly rely on unsupervised objectives, such as language modeling (Shen et al., 2018b; 2019; Kim et al., 2019a;b) and cloze-style word prediction (Drozdov et al., 2019) to train their task-oriented models. On the other hand, there is a trend in the natural language processing (NLP) community of leveraging pre-trained language models (LMs), e.g., ELMo (Peters et al., 2018) and BERT (Devlin et al., 2019), as a means of acquiring contextualized word representations. These representations have proven to be surprisingly effective, playing key roles in recent improvements in various models for diverse NLP tasks.

In this paper, inspired by the fact that the training objectives of both the approaches for grammar induction and for training LMs are identical, namely, (masked) language modeling, we investigate whether pre-trained LMs can also be utilized for grammar induction/unsupervised parsing, especially *without* training. Specifically, we focus on extracting constituency trees from pre-trained LMs without fine-tuning or introducing another task-specific module, at least one of which is usually required in other cases where representations from pre-trained LMs are employed. This restriction provides us with some advantages: (1) it enables us to derive strong baselines for grammar induction with reduced time and space complexity, offering a chance to reexamine the current status of existing grammar induction methods, (2) it facilitates an analysis on how much and what kind of syntactic information each pre-trained LM contains in its intermediate representations and attention distributions in terms of phrase-structure grammar, and (3) it allows us to easily inject biases into our framework, for instance, to encourage the right-skewness of the induced trees, resulting in performance gains in English unsupervised parsing.

First, we briefly mention related work (§2). Then, we introduce the intuition behind our proposal in detail (§3), which is motivated by our observation that we can cluster words in a sentence according to the similarity of their attention distributions over words in the sentence. Based on this intuition, we define a straightforward yet effective method (§4) of drawing constituency trees directly from pre-trained LMs with no fine-tuning or addition of task-specific parts, instead resorting to the concept of *Syntactic Distance* (Shen et al., 2018a;b). Then, we conduct experiments (§5) on the induced constituency trees, discovering some intriguing phenomena. Moreover, we analyze the pre-trained

LMs and constituency trees from various points of view, including looking into which layer(s) of the LMs is considered to be sensitive to phrase information (§6).

To summarize, our contributions in this work are as follows:

- By investigating the attention distributions from Transformer-based pre-trained LMs, we show that there is evidence to suggest that several attention heads of the LMs exhibit syntactic structure akin to constituency grammar.

- Inspired by the above observation, we propose a method that facilitates the derivation of constituency trees from pre-trained LMs without training. We also demonstrate that the induced trees can serve as a strong baseline for English grammar induction.

- We inspect, in view of our framework, what type of syntactic knowledge the pre-trained LMs capture, discovering interesting facts, e.g., that some pre-trained LMs are more aware of adverb phrases than other approaches.

## 2  RELATED WORK

Grammar induction is a task whose goal is to infer from sequential data grammars which generalize, and are able to account for unseen data (Lari & Young (1990); Clark (2001); Klein & Manning (2002; 2004), to name a few). Traditionally, this was done by learning explicit grammar rules (e.g., context free rewrite rules), though more recent methods employ neural networks to learn such rules implicitly, focusing more on the induced grammars' ability to generate or parse sequences.

Specifically, Shen et al. (2018b) proposed Parsing-Reading-Predict Network (PRPN) where the concept of *Syntactic Distance* is first introduced. They devised a neural model for language modeling where the model is encouraged to recognize syntactic structure. The authors also probed the possibility of inducing constituency trees without access to gold-standard trees by adopting an algorithm that recursively splits a sequence of words into two parts, the split point being determined according to correlated syntactic distances; the point having the biggest distance becomes the first target of division. Shen et al. (2019) presented a model called Ordered Neurons (ON), which is a revised version of LSTM (Long Short-Term Memory, Hochreiter & Schmidhuber (1997)) which reflects the hierarchical biases of natural language and can be used to compute syntactic distances. Shen et al. (2018a) trained a supervised parser relying on the concept of syntactic distance.

Other studies include Drozdov et al. (2019), who trained deep inside-outside recursive autoencoders (DIORA) to derive syntactic trees in an exhaustive way with the aid of the inside-outside algorithm, and Kim et al. (2019a) who proposed Compound Probabilistic Context-Free Grammars (compound PCFG), showing that neural PCFG models are capable of producing promising unsupervised parsing results. Li et al. (2019) proved that an ensemble of unsupervised parsing models can be beneficial, while Shi et al. (2019) utilized additional training signals from pictures related with input text. Dyer et al. (2016) proposed Recurrent Neural Network Grammars (RNNG) for both language modeling and parsing, and Kim et al. (2019b) suggested an unsupervised variant of the RNNG. There also exists another line of research on task-specific latent tree learning (Yogatama et al., 2017; Choi et al., 2018; Havrylov et al., 2019; Maillard et al., 2019). The goal here is not to construct linguistically plausible trees, but to induce trees fitted to improving target performance. Naturally, the induced performance-based trees need not resemble linguistically plausible trees, and some studies (Williams et al., 2018a; Nangia & Bowman, 2018) examined the apparent fact that performance-based and lingusitically plausible trees bear little resemblance to one another.

Concerning pre-trained language models (Peters et al. (2018); Devlin et al. (2019); Radford et al. (2019); Yang et al. (2019); Liu et al. (2019b), *inter alia*)—particularly those employing a Transformer architecture (Vaswani et al., 2017)—these have proven to be helpful for diverse NLP downstream tasks. In spite of this, there is no vivid picture for explaining what particular factors contribute to performance gains, even though some recent work has attempted to shed light on this question. In detail, one group of studies (Raganato & Tiedemann (2018); Clark et al. (2019); Ethayarajh (2019); Hao et al. (2019); Voita et al. (2019), *inter alia*) has focused on dissecting the intermediate representations and attention distributions of the pre-trained LMs, while the another group of publications (Mareček & Rosa (2018); Goldberg (2019); Hewitt & Manning (2019); Liu et al. (2019a); Rosa & Mareček (2019), to name a few) delve into the question of the existence of syntactic knowledge in

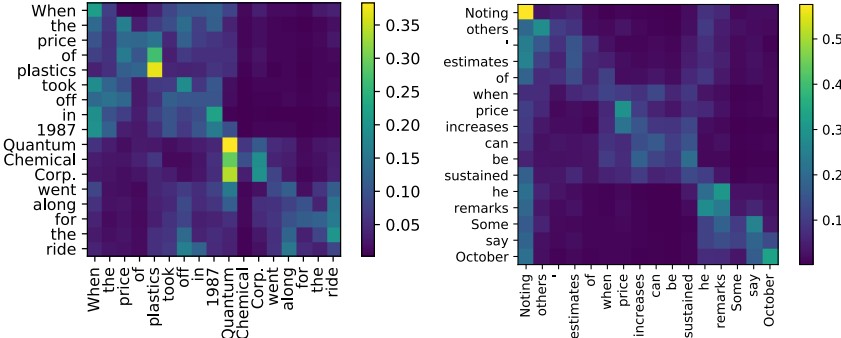

Figure 1: Self-attention heatmaps from two different pre-trained LMs. (Left) A heatmap for the average of attention distributions from the 7th layer of the XLNet-base (Yang et al., 2019) model given the sample sentence. (Right) A heatmap for the average of attention distributions from the 9th layer of the BERT-base (Devlin et al., 2019) model given another sample sentence. We can easily spot the chunks of words on the two heatmaps that are correlated with the constituents of the input sentences, e.g., (Left) 'the price of plastics', 'took off in 1987', 'Quantum Chemical Corp.', (Right) 'when price increases can be sustained', and 'he remarks'.

Transformer-based models. Particularly, Mareček & Rosa (2019) proposed an algorithm for extracting constituency trees from Transformers trained for machine translation, which is similar to our approach.

## 3   MOTIVATION

As pioneers in the literature have pointed out, the multi-head self-attention mechanism (Vaswani et al., 2017) is a key component in Transformer-based language models, and it seems this mechanism empowers the models to capture certain semantic and syntactic information existing in natural language. Among a diverse set of knowledge they may capture, in this work we concentrate on phrase-structure grammar by seeking to extract constituency trees directly from their attention information and intermediate weights.

In preliminary experiments, where we manually visualize and investigate the intermediate representations and attention distributions of several pre-trained LMs given input, we have found some evidence which suggests that the pre-trained LMs exhibit syntactic structure akin to constituency grammar in some degree. Specifically, we have noticed some patterns which are often displayed in self-attention heatmaps as explicit horizontal lines, or groups of rectangles of various sizes. As an attention distribution of a word in an input sentence corresponds to a row in a heatmap matrix, we can say that the appearance of these patterns indicates the existence of groups of words where the attention distributions of the words in the same group are relatively similar. Interestingly, we have also discovered the fact that the groups of words we observed are fairly correlated with the constituents of the input sentence, as shown in Figure 1 (above) and Figure 3 (in Appendix A.1).

Even though we have identified some patterns which match with the constituents of sentences, it is not enough to conclude that the pre-trained LMs are aware of syntactic phrases as found in phrase-structure grammars. To demonstrate the claim, we attempt to obtain constituency trees in a zero-shot learning fashion, relying only on the knowledge from the pre-trained LMs. To this end, we suggest the following, inspired from our finding: two words in a sentence are syntactically *close* to each other (i.e., the two words belong to the same constituent) if their attention distributions over words in the sentence are also *close* to each other. Note that this implicitly presumes that each word is more likely to attend more on the words in the same constituent to enrich its representation in the pre-trained LMs. Finally, we utilize the assumption to compute syntactic distances between each pair of adjacent words in a sentence, from which the corresponding constituency tree can be built.

## 4 PROPOSED METHOD

### 4.1 SYNTACTIC DISTANCE AND TREE CONSTRUCTION

We leverage the concept of *Syntactic Distance* proposed by Shen et al. (2018a;b) to draw constituency trees from raw sentences in an intuitive way. Formally, given a sequence of words in a sentence, $w_1, w_2, \ldots, w_n$, we compute $\mathbf{d} = [d_1, d_2, \ldots, d_{n-1}]$ where $d_i$ corresponds to the syntactic distance between $w_i$ and $w_{i+1}$. Each $d_i$ is defined as follows:

$$d_i = f(g(w_i), g(w_{i+1})), \tag{1}$$

where $f(\cdot, \cdot)$ and $g(\cdot)$ are a distance measure function and representation extractor function, respectively. The function $g$ converts each word into the corresponding vector representation, while $f$ computes the syntactic distance between the two words given their representations. Once $\mathbf{d}$ is derived, it can be easily converted into the target constituency tree by a simple algorithm following Shen et al. (2018a).[1] For details of the algorithm, we refer the reader to Appendix A.2.

Although previous studies attempted to explicitly train the functions $f$ and $g$ with supervision (with access to gold-standard trees, Shen et al. (2018a)) or to obtain them as a by-product of training particular models that are carefully designed to recognize syntactic information (Shen et al., 2018b; 2019), in this work we stick to simple distance metric functions for $f$ and pre-trained LMs for $g$, forgoing any training process. In other words, we focus on investigating the possibility of pre-trained LMs possessing constituency information in a form that can be readily extracted with straightforward computations. If the trees induced by the syntactic distances derived from the pre-trained LMs are similar enough to gold-standard syntax trees, we can reasonably claim that the LMs resemble phrase-structure.

### 4.2 PRE-TRAINED LANGUAGE MODELS

We consider four types of recently proposed language models. These are: BERT (Devlin et al., 2019), GPT-2 (Radford et al., 2019), RoBERTa (Liu et al., 2019b), and XLNet (Yang et al., 2019). They all have in common that they are based on the Transformer architecture and have been proven to be effective in natural language understanding (Wang et al., 2019) or generation. We handle two variants for each LM, varying in the number of layers, attention heads, and hidden dimensions, resulting in eight different cases in total. In particular, each LM has two variants. (1) base: consists of $l$=12 layers, $a$=12 attention heads, and $d$=768 hidden dimensions, while (2) large: has $l$=24 layers, $a$=16 attention heads, and $d$=1024 hidden dimensions.[2] We deal with a wide range of pre-trained LMs, unlike previous work which has mostly analyzed a specific model, particularly BERT. For details about each LM, we refer readers to the respective original papers.

In terms of our formulation, each LM instance provides two categories of representation extractor functions, $G^v$ and $G^d$. Specifically, $G^v$ refers to a set of functions $\{g^v_j | j = 1, \ldots, l\}$, each of which simply outputs the intermediate hidden representation of a given word on the $j$th layer of the LM. Likewise, $G^d$ is a set of functions $\{g^d_{(j,k)} | j = 1, \ldots, l, k = 1, \ldots, a + 1\}$, each of which outputs the attention distribution of an input word by the $k$th attention head on the $j$th layer of the LM. Even though our main motivation comes from the self-attention mechanism, we also deal with the intermediate hidden representations present in the pre-trained LMs by introducing $G^v$, considering that the hidden representations serve as storage of collective information taken from the processing of the pre-trained LMs. Note that $k$ ranges up to $a + 1$, not $a$, implying that we consider the average of all attention distributions on the same layer in addition to the individual ones. This averaging function can be regarded as an ensemble of other functions in the layer which are specialized for different aspects of information, and we expect that this technique will provide a better option in some cases as reported in previous work (Li et al., 2019).

---

[1] Our parsing algorithm is an unbiased method in contrast to one (named as $\overline{\text{COO}}$ parser by Dyer et al. (2019)) employed in most previous studies (Shen et al., 2018b; 2019; Htut et al., 2018; Li et al., 2019; Shi et al., 2019). This choice enables us to investigate the exact extent to which pre-trained LMs contribute to their performance on unsupervised parsing, considering the fact revealed recently by Dyer et al. (2019) that the $\overline{\text{COO}}$ parser has potential issues that it prefers right-branching trees and does not cover all possible tree derivations. Furthermore, we can directly adjust the right-branching bias using our method in Section 4.4 if needed.

[2] In case of GPT-2, 'GPT2' corresponds to the 'base' variant while 'GPT2-medium' to the 'large' one.

One remaining issue is that all the pre-trained LMs we use regard each input sentence as a sequence of *subword* tokens, while our formulation assumes words cannot be further divided into smaller tokens. To resolve this difference, we tested certain heuristics that guide how subword tokens for a complete word should be exploited to represent the word, and we have empirically found that the best result comes when each word is represented by an average of the representations of its subwords.[3] Therefore, we adopt the above heuristic in this work for cases where a word is tokenized into more than two parts.

### 4.3 DISTANCE MEASURE FUNCTIONS

For the distance measure function $f$, we prepare three options ($F^v$) for $G^v$ and two options ($F^d$) for $G^d$. Formally, $f \in F^v \cup F^d$, where $F^v = \{\text{COS}, \text{L1}, \text{L2}\}$, $F^d = \{\text{JSD}, \text{HEL}\}$. COS, L1, L2, JSD, and HEL correspond to Cosine, L1, and L2, Jensen-Shannon, and Hellinger distance respectively. The functions in $F^v$ are only compatible with the elements of $G^v$, and the same holds for $F^d$ and $G^d$. The exact definition of each function is listed in Appendix A.3.

### 4.4 INJECTING BIAS INTO SYNTACTIC DISTANCES

One of the main advantages we obtain by leveraging syntactic distances to derive parse trees is that we can easily inject inductive bias into our framework by simply modifying the values of the syntactic distances. Hence, we investigate whether the extracted trees from our method can be further refined with the aid of additional biases. To this end, we introduce a well-known bias for English constituency trees—the *right-skewness* bias—in a simple linear form.[4] Namely, our intention is to influence the induced trees such that they are moderately right-skewed following the nature of gold-standard parse trees in English.

Formally, we compute $\hat{d}_i$ by appending the following linear bias term to every $d_i$:

$$\hat{d}_i = d_i + \lambda \cdot \mathbf{AVG}(\mathbf{d}) \times (1 - 1/(m-1) \times (i-1)), \tag{2}$$

where $\text{AVG}(\cdot)$ outputs an average of all elements in a vector, $\lambda$ is a hyperparameter, and $i$ ranges from 1 to $m = n - 1$. We write $\hat{\mathbf{d}} = [\hat{d}_1, \hat{d}_2, \ldots, \hat{d}_m]$ in place of $\mathbf{d}$ to signify biased syntactic distances.

The main purpose of introducing such a bias is examining what changes are made to the resulting tree structures rather than boosting quantitative performance *per se*, though it is of note that it serves this purpose as well. We believe that this additional consideration is necessary based on two points. First, English is what is known as a head-initial language. That is, given a selector and argument, the selector has a strong tendency to appear on the left, e.g., 'eat food', or 'to Canada'. Head-initial languages therefore have an in-built preference for right-branching structures. By adjusting the bias injected into syntactic distances derived from pre-trained LMs, we can figure out whether the LMs are capable of inducing the right-branching bias, which is one of the main properties of English syntax; if injecting the bias does not influence the performance of the LMs on unsupervised parsing, we can conjecture they are inherently capturing the bias to some extent. Second, as mentioned before, we have witnessed some previous work (Shen et al., 2018b; 2019; Htut et al., 2018; Li et al., 2019; Shi et al., 2019) where the right-skewness bias is implicitly exploited, although it could be regarded as not ideal. What we intend to focus on is the question about which benefits the bias provides for such parsing models, leading to overall performance improvements. In other words, we look for what the exact contribution of the bias is when it is injected into grammar induction models, by explicitly controlling the bias using our framework.

---

[3]We also tried other heuristics following previous work (Kitaev & Klein, 2018), e.g., using the first or last subword of a word as representative, but this led to no performance gains.

[4]It is necessary to carefully design biases for other languages as they have their own properties.

## 5 EXPERIMENTS

### 5.1 GENERAL SETTINGS

#### 5.1.1 DATASETS

In this section, we conduct unsupervised constituency parsing on two datasets. The first dataset is WSJ Penn Treebank (PTB, Marcus et al. (1993)), in which human-annotated gold-standard trees are available. We use the standard split of the dataset—2-21 for training, 22 for validation, and 23 for test. The second one is MNLI (Williams et al., 2018b), which is originally designed to test natural language inference but often utilized as a means of evaluating parsers. It contains constituency trees produced by an external parser (Klein & Manning, 2003). We leverage the union of two different versions of the MNLI development set as test data following convention (Htut et al., 2018; Drozdov et al., 2019), and we call it the MNLI test set in this paper. Moreover, we randomly sample 40K sentences from the training set of the MNLI to utilize them as a validation set. To preprocess the datasets, we follow the setting of Kim et al. (2019a) with the minor exceptions that words are not lower-cased and number characters are preserved instead of being substituted by a special character.

#### 5.1.2 IMPLEMENTATION DETAILS

For implementation, to compare pre-trained LMs in an unified manner, we resort to an integrated PyTorch codebase that supports all the models we consider.[5] For each LM, we tune the best combination of $f$ and $g$ functions using the validation set. Then, we derive a set of $\mathbf{d}$ for sentences in the test set using the chosen functions, followed by the resulting constituency trees converted from each $\mathbf{d}$ by the tree construction algorithm in Section 4.1. In addition to sentence-level F1 (S-F1) score, we report label recall scores for six main categories: SBAR, NP, VP, PP, ADJP, and ADVP. We also present the results of utilizing $\hat{\mathbf{d}}$ instead of $\mathbf{d}$, empirically setting the bias hyperparameter $\lambda$ as 1.5. We do not fine-tune the LMs on domain-specific data, as we here focus on finding their universal characteristics.

We take four naïve baselines into account, random (averaged over 5 trials), balanced, left-branching, and right-branching binary trees. In addition, we present two more baselines which are identical to our models except that their $g$ functions are based on a randomly initialized XLNet-base rather than pre-trained ones. To be concrete, We provide 'Random XLNet-base ($F^v$)' which applies the functions in $F^v$ on random hidden representations and 'Random XLNet-base ($F^d$)' that utilizes the functions in $F^d$ and random attention distributions, respectively. Considering the randomness of initialization and possible choices for $f$, the final score for each of the baselines is calculated as an average over 5 trials of each possible $f$, i.e., an average over $5 \times 3$ runs in case of $F^v$ and $5 \times 2$ runs for $F^d$. These baselines enable us to estimate the exact advantage we obtain by pre-training LMs, effectively removing additional unexpected gains that may exist. Furthermore, we compare our parse trees against ones from existing grammar induction models.

All scripts used in our experiments will be publicly available for reproduction and further analysis.[6]

### 5.2 EXPERIMENTAL RESULTS ON PTB

In Table 1, we report the results of the various models on the PTB test set. First of all, our method combined with pre-trained LMs shows competitive or comparable results in terms of S-F1 even without the right-skewness bias. This result implies that the extracted trees from our method can be regarded as a baseline for English grammar induction. Moreover, pre-trained LMs show substantial improvements over Random Transformers (XLNet-base), demonstrating that training language models on large corpora, in fact, enables the LMs to be more aware of syntactic information.

When the right-skewness bias is applied to syntactic distances derived from pre-trained LMs, the S-F1 scores of the LMs increase by up to ten percentage points. This improvement indicates that the pre-trained LMs do not properly capture the largely right-branching nature of English syntax, at least when observed through the lens of our framework. By explicitly controlling the bias through our framework and observing the performance gap between our models with and without the bias,

---

[5] https://github.com/huggingface/transformers
[6] https://github.com/galsang/trees_from_transformers

Table 1: Results on the PTB test set. Bold numbers correspond to the top 3 results for each column. L: layer number, A: attention head number (AVG: the average of all attentions). †: Results reported by Kim et al. (2019a). ‡: Approaches in which $\overline{\text{COO}}$ parser is utilized.

| Model | $f$ | L | A | S-F1 | SBAR | NP | VP | PP | ADJP | ADVP |
|---|---|---|---|---|---|---|---|---|---|---|
| **Baselines** | | | | | | | | | | |
| Random Trees | - | - | - | 18.1 | 8% | 23% | 12% | 18% | 23% | 28% |
| Balanced Trees | - | - | - | 18.5 | 7% | 27% | 8% | 18% | 27% | 25% |
| Left Branching Trees | - | - | - | 8.7 | 5% | 11% | 0% | 5% | 2% | 8% |
| Right Branching Trees | - | - | - | 39.4 | **68%** | 24% | **71%** | 42% | 27% | 38% |
| Random XLNet-base ($F^v$) | - | - | - | 19.6 | 9% | 26% | 12% | 20% | 23% | 24% |
| Random XLNet-base ($F^d$) | - | - | - | 20.1 | 11% | 25% | 14% | 19% | 22% | 26% |
| **Pre-trained LMs (w/o bias)** | | | | | | | | | | |
| BERT-base | JSD | 9 | AVG | 32.4 | 28% | 42% | 28% | 31% | 35% | 63% |
| BERT-large | HEL | 17 | AVG | 34.2 | 34% | 43% | 27% | 39% | 37% | 57% |
| GPT2 | JSD | 9 | 1 | 37.1 | 32% | 47% | 27% | 55% | 27% | 36% |
| GPT2-medium | JSD | 10 | 13 | 39.4 | 41% | 51% | 21% | **67%** | 33% | 44% |
| RoBERTa-base | JSD | 9 | 4 | 33.8 | 40% | 38% | 33% | 43% | 42% | 57% |
| RoBERTa-large | JSD | 14 | 5 | 34.1 | 29% | 46% | 30% | 37% | 28% | 40% |
| XLNet-base | HEL | 9 | AVG | 40.1 | 35% | 56% | 26% | 38% | 47% | 68% |
| XLNet-large | L2 | 11 | - | 38.1 | 36% | 51% | 26% | 41% | 45% | **69%** |
| **Pre-trained LMs (w/ bias λ=1.5)** | | | | | | | | | | |
| BERT-base | HEL | 9 | AVG | 42.3 | 45% | 46% | **49%** | 43% | 41% | 65% |
| BERT-large | HEL | 17 | AVG | 44.4 | 55% | 48% | 48% | 52% | 41% | 62% |
| GPT2 | JSD | 9 | 1 | 41.3 | 43% | 49% | 38% | **58%** | 27% | 43% |
| GPT2-medium | HEL | 2 | 1 | 42.3 | 54% | 50% | 39% | 56% | 24% | 41% |
| RoBERTa-base | JSD | 8 | AVG | 42.1 | 51% | 44% | 44% | 55% | 40% | 66% |
| RoBERTa-large | JSD | 12 | AVG | 42.3 | 40% | 53% | 44% | 44% | **48%** | 56% |
| XLNet-base | HEL | 7 | AVG | **48.3** | **62%** | 53% | **50%** | **58%** | **49%** | **74%** |
| XLNet-large | HEL | 11 | AVG | 46.7 | 57% | 50% | 54% | 50% | **57%** | **73%** |
| **Other models** | | | | | | | | | | |
| PRPN(tuned)[†‡] | - | - | - | 47.3 | 50% | 59% | 46% | 57% | 44% | 32% |
| ON(tuned)[†‡] | - | - | - | 48.1 | 51% | **64%** | 41% | 54% | 38% | 31% |
| Neural PCFG[†] | - | - | - | **50.8** | 52% | **71%** | 33% | **58%** | 32% | 45% |
| Compound PCFG[†] | - | - | - | **55.2** | **56%** | **74%** | 41% | **68%** | 40% | 52% |

we confirm that the main contribution of the bias comes from its capability to capture subordinate clauses (SBAR) and verb phrases (VP). This observation provides a hint for what some previous work on unsupervised parsing desired to obtain by introducing the bias to their models. It is intriguing to see that all of the existing grammar induction models are inferior to the right-branching baseline in recognizing SBAR and VP (although some of them already utilized the right-skewness bias), implying that the same problem—models do not properly capture the right-branching nature—may also exist in current grammar induction models. One possible assumption is that the models do not need the bias to perform well in language modeling, although future work should provide a rigorous analysis about the phenomenon.

On the other hand, the existing models show exceptionally high recall scores on noun phrases (NP), even though our pre-trained LMs also have success to some extent in capturing noun phrases compared to naïve baselines. From this, we conjecture that neural models trained with a language modeling objective become largely equipped with the ability to understand the concept of NP. In contrast, the pre-trained LMs record the best recall scores on adjective and adverb phrases (ADJP and ADVP), suggesting that the LMs and existing models capture disparate aspects of English syntax to differing degrees. To further explain why some pre-trained LMs are good at capturing ADJPs and ADVPs, we manually investigated the attention heatmaps of the sentences that contain ADJPs or ADVPs. From the inspection, we empirically found that there are some keywords—including 'two', 'ago', 'too', and 'far'—which have different patterns of attention distributions compared to those of their neighbors and that these keywords can be a clue for our framework to recognize the existence of ADJPs or ADJPs. It is also worth mentioning that ADJPs and ADVPs consist of a relatively smaller number of words than those of SBAR and VP, indicating that the LMs combined with our method have strength in correctly finding small chunks of words, i.e., low-level phrases.

Meanwhile, in comparison with other LM models, GPT-2 and XLNet based models demonstrate their effectiveness and robustness in unsupervised parsing. Particularly, the XLNet-base model serves as a robust baseline achieving the top performance among LM candidates. One plausible explanation for this outcome is that the training objective of XLNet, which considers both autoencoding (AE) and autoregressive (AR) features, might encourage the model to be better aware of

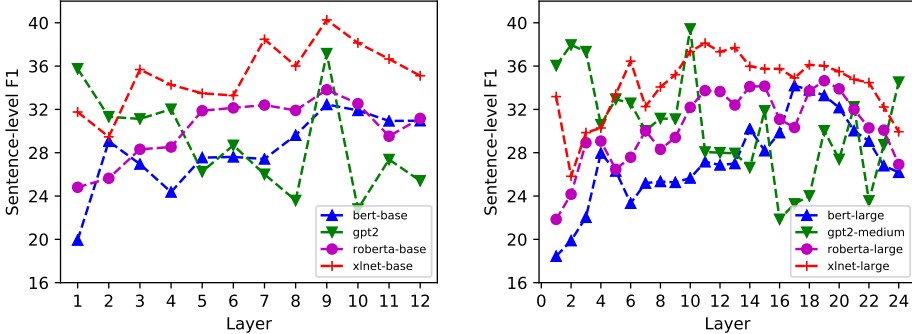

Figure 2: The best layer-wise S-F1 scores of each LM instance on the PTB test set. (Left) The performance of the X-'base' models. (Right) The performance of the X-'large' models.

phrase structure than other LMs. Another possible hypothesis is that AR objective functions (e.g., typical language modeling) are more effective in training syntax-aware neural models than AE objectives (e.g., masked language modeling), as both GPT-2 and XLNet are pre-trained on AR variants. However, it is hard to conclude what factors contribute to their high performance at this stage.

Interestingly, there is an obvious trend that the functions in $F^d$—the distance measure functions for attention distributions—lead most of the LM instances to the best parsing results, indicating that deriving parse trees from attention information can be more compact and efficient than extracting them from the LMs' intermediate representations, which should contain linguistic knowledge beyond phrase structure. In addition, the results in Table 1 show that large parameterizations of the LMs generally increase their parsing performance, but this improvement is not always guaranteed. Meanwhile, as we expected in Section 4.2 and as seen in the 'A' (attention head number) column of Table 1, the average of attention distributions in the same layer often provides better results than individual attention distributions.

## 5.3 EXPERIMENTAL RESULTS ON MNLI

We present the results of various models on the MNLI test set in Table 3 of Appendix A.5. We observe trends in the results which mainly coincide with those of the PTB dataset. Particularly, (1) right-branching trees are strong baselines for the task, especially showing their strengths in capturing SBAR and VP clauses/phrases, (2) our method resorting to the LM instances is also comparable to the right-branching trees, demonstrating its superiority in recognizing different aspects of phrase categories including prepositional phrases (PP) and adverb phrases (ADVP), and (3) attention distributions seem more effective for distilling the phrase structures of sentences than intermediate representations.

However, there are some issues worth mentioning. First, the right-branching baseline seems to be even stronger in the case of MNLI, recording a score of over 50 in sentence-level F1. We conjecture that this result comes principally from two reasons: (1) the average length of sentences in MNLI is much shorter than in PTB, giving a disproportionate advantage to naïve baselines, and (2) our data preprocessing, which follows Kim et al. (2019a), removes all punctuation marks, unlike previous work (Htut et al., 2018; Drozdov et al., 2019), leading to an unexpected advantage for the right-branching scheme. Moreover, it deserves to consider the fact that the gold-standard parse trees in MNLI are not human-annotated, rather automatically generated.

Second, in terms of consistency in identifying the best choice of $f$ and $g$ for each LM, we observe that most of the best combinations of $f$ and $g$ tuned for PTB do not correspond well to the best ones for MNLI. Does this observation imply that a specific combination of these functions and the resulting performance do not generalize well across different data domains? To clarify, we manually investigated the performance of some combinations of $f$ and $g$, which are tuned on PTB but tested on MNLI instead. As a result, we discover that particular combinations of $f$ and $g$ which are good at PTB are also competitive on MNLI, even though they fail to record the best scores on MNLI.

Table 2: Results of training a pseudo-optimum $f_{\text{ideal}}$ with PTB and XLNet-base model.

| Model | $f$ | L | A | S-F1 | SBAR | NP | VP | PP | ADJP | ADVP |
|---|---|---|---|---|---|---|---|---|---|---|
| **Baselines (from Table 1)** | | | | | | | | | | |
| Random XLNet-base ($F^v$) | - | - | - | 19.6 | 9% | 26% | 12% | 20% | 23% | 24% |
| Random XLNet-base ($F^d$) | - | - | - | 20.1 | 11% | 25% | 14% | 19% | 22% | 26% |
| XLNet-base ($\lambda=0$) | JSD | 9 | AVG | 40.1 | 35% | 56% | 26% | 38% | 47% | 68% |
| XLNet-base ($\lambda=1.5$) | HEL | 7 | AVG | 48.3 | **62%** | 53% | 50% | 58% | 49% | **74%** |
| **Trained models (w/ gold trees)** | | | | | | | | | | |
| Random XLNet-base | $f_{\text{ideal}}$ | - | - | 41.2 | 28% | 58% | 29% | 50% | 35% | 41% |
| XLNet-base (worst case) | $f_{\text{ideal}}$ | 1 | - | 58.0 | 47% | 75% | 56% | 71% | 50% | 61% |
| XLNet-base (best case) | $f_{\text{ideal}}$ | 7 | - | **65.1** | 61% | **82%** | **67%** | **78%** | **55%** | 73% |

Concretely, the union of $f^d$(JSD) and $g_{(9,13)}^d$—the best duo for the XLNet-base on PTB—achieves 39.2 in sentence-level F1 on MNLI, which is very close to the top performance (39.3) we can obtain when leveraging the XLNet-base. It is also worth noting that GPT-2 and XLNet are efficient in capturing PP and ADVP respectively, regardless of the data domain and the choice of $f$ and $g$.

# 6 FURTHER ANALYSIS

## 6.1 PERFORMANCE COMPARISON BY LAYER

To take a closer look at how different the layers of the pre-trained LMs are in terms of parsing performance, we retrieve the best sentence-level F1 scores from the $l$th layer of an LM from all combinations of $f$ and $g_l$, with regard to the PTB and MNLI respectively. Then we plot the scores as graphs in Figure 2 for the PTB and Figure 4 in Appendix A.4 for the MNLI. Each score is from the models to which the bias is *not* applied.

From the graphs, we observe several patterns. First, XLNet-based models outperform other competitors across most of the layers. Second, the best outcomes are largely shown in the middle layers of the LMs akin to the observation from Shen et al. (2019), except for some cases where the first layers (especially in case of MNLI) record the best. Interestingly, GPT-2 shows a decreasing trend in its output values as the layer becomes high, while other models generally exhibit the opposite pattern. Moreover, we discover from raw statistics that regardless of the choice of $f$ and $g_l$, the parsing performance reported as S-F1 is moderately correlated with the layer number $l$. In other words, it seems that there are some particular layers in the LMs which are more sensitive to syntactic information.

## 6.2 ESTIMATING THE UPPER LIMIT OF DISTANCE MEASURE FUNCTIONS

Although we have introduced effective candidates for $f$, we explore the potential of extracting more sophisticated trees from pre-trained LMs, supposing we are equipped with a pseudo-optimum $f$, call it $f_{\text{ideal}}$. To obtain $f_{\text{ideal}}$, we train a simple linear layer on each layer of the pre-trained LMs *with supervision* from the gold-standard trees of the PTB training set, while $g$ remains unchanged—the pre-trained LMs are frozen during training. We choose the XLNet-base model as a representative for the pre-trained LMs. For more details about experimental settings, refer to Appendix A.6.

In Table 2, we present three new results using $f_{\text{ideal}}$. As a baseline, we report the performance of $f_{\text{ideal}}$ with a randomly initialized XLNet-base. Then, we list the worst and best result of $f_{\text{ideal}}$ according to $g$, when it is combined with the pre-trained LM. We here mention some findings from the experiment. First, comparing the results with the pre-trained LM against one with the random LM, we reconfirm that pre-training an LM apparently enables the model to capture some aspects of grammar. Specifically, our method is comparable to the linear model trained on the gold-standard trees. Second, we find that there is a tendency for the performance of $f_{\text{ideal}}$ relying on different LM layers to follow one we already observed in Section 6.1—the best result comes from the middle layers of the LM while the worst from the first and last layer. Third, we identify that the LM has a potential to show improved performance on grammar induction by adopting a more sophisticated $f$. However, we emphasize that our method equipped with a simple $f$ without gold-standard trees is remarkably reasonable in recognizing constituency grammar, being especially good at catching ADJP and ADVP.

### 6.3 Constituency Tree Examples

We visualize several gold-standard trees from PTB and the corresponding tree predictions for comparison. For more details, we refer readers to Appendix A.7.

## 7 Conclusions and Future Work

In this paper, we propose a simple but effective method of inducing constituency trees from pre-trained language models in a zero-shot learning fashion. Furthermore, we report a set of intuitive findings observed from the extracted trees, demonstrating that the pre-trained LMs exhibit some properties similar to constituency grammar. In addition, we show that our method can serve as a strong baseline for English grammar induction when combined with (or even without) appropriate linguistic biases.

On the other hand, there are still remaining issues that can be good starting points for future work. First, although we analyzed our method based on two popular datasets, we focused only on English grammar induction. As each language has its own properties (and correspondingly would need individualized biases), it is desirable to expand this work to other languages. Second, it would also be desirable to investigate whether further improvements can be achieved by directly grafting the pre-trained LMs onto existing grammar induction models. Lastly, by verifying the usefulness of the knowledge from the pre-trained LMs and linguistic biases for grammar induction, we want to point out that there is still much room for improvement in the existing grammar induction models.

### Acknowledgments

We would like to thank Reinald Kim Amplayo and the anonymous reviewers for their thoughtful and valuable comments. This work was supported by the National Research Foundation of Korea (NRF) grant funded by the Korea government (MSIT) (NRF2016M3C4A7952587).

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

# A APPENDIX

## A.1 ATTENTION HEATMAP EXAMPLES

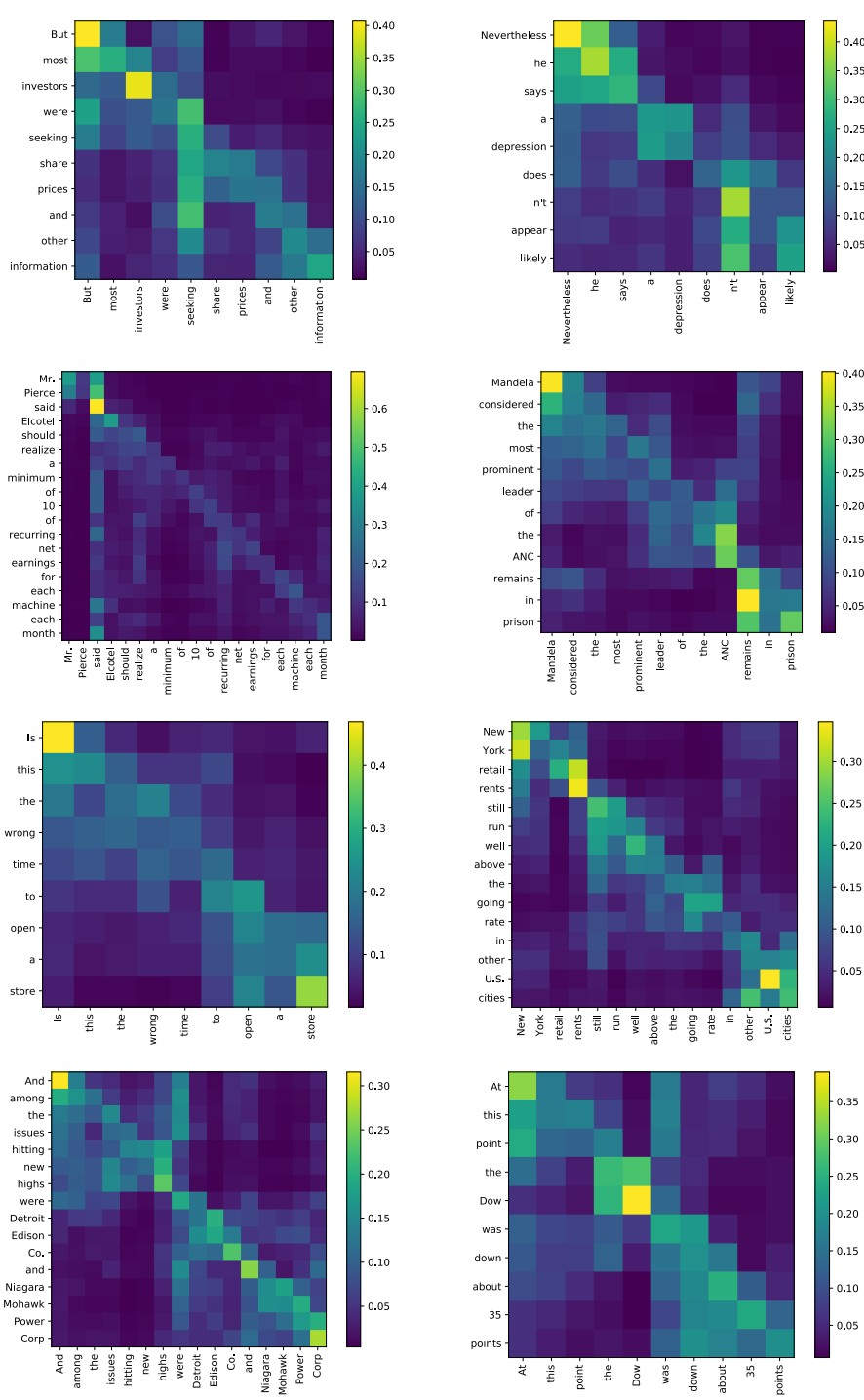

Figure 3: Self-attention heatmaps for the average of all attention distributions from the 7th layer of the XLNet-base model, given a set of input sentences.

## A.2 Tree Construction Algorithm with Syntactic Distances

**Algorithm 1** Syntactic Distances to Binary Constituency Tree (originally from Shen et al. (2018a))

1: $S = [w_1, w_2, \ldots, w_n]$: a sequence of words in a sentence of length $n$.
2: $\mathbf{d} = [d_1, d_2, \ldots, d_{n-1}]$: a vector whose elements are the distances between every two adjacent words.
3: **function** TREE$(S, \mathbf{d})$
4:     **if** $\mathbf{d} = []$ **then**
5:         node $\leftarrow$ Leaf$(S[0])$
6:     **else**
7:         $i \leftarrow \arg\max_i(\mathbf{d})$
8:         child$_l \leftarrow$ TREE$(S_{\leq i}, \mathbf{d}_{<i})$
9:         child$_r \leftarrow$ TREE$(S_{>i}, \mathbf{d}_{>i})$
10:       node $\leftarrow$ Node(child$_l$, child$_r$)
11:     **end if**
12:     **return** node
13: **end function**

## A.3 Distance Measure Functions

Table 3: The definitions of distance measure functions for computing syntactic distances between two adjacent words in a sentence. Note that $\mathbf{r} = g^v(w_i)$, $\mathbf{s} = g^v(w_{i+1})$, $P = g^d(w_i)$, and $Q = g^d(w_{i+1})$, respectively. $d$: hidden embedding size, $n$: the number of words $(w)$ in a sentence $(S)$.

| Function $(f)$ | Definition |
|---|---|
| Functions for intermediate representations $(F^v)$ | |
| Cos$(\mathbf{r}, \mathbf{s})$ | $\left(\mathbf{r}^\top \mathbf{s} / \left((\sum_{i=1}^d r_i^2)^{\frac{1}{2}} \cdot (\sum_{i=1}^d s_i^2)^{\frac{1}{2}}\right) + 1\right)/2$ |
| L1$(\mathbf{r}, \mathbf{s})$ | $\sum_{i=1}^d |r_i - s_i|$ |
| L2$(\mathbf{r}, \mathbf{s})$ | $(\sum_{i=1}^d (r_i - s_i)^2)^{\frac{1}{2}}$ |
| Functions for attention distributions $(F^d)$ | |
| JSD$(P\|Q)$ | $((D_{\text{KL}}(P\|M) + D_{\text{KL}}(Q\|M))/2)^{\frac{1}{2}}$ where $M = (P + Q)/2$ and $D_{\text{KL}}(A\|B) = \sum_{w \in S} A(w) \log(A(w)/B(w))$ |
| HEL$(P, Q)$ | $\frac{1}{\sqrt{2}}(\sum_{i=1}^n (\sqrt{p_i} - \sqrt{q_i})^2)^{\frac{1}{2}}$ |

## A.4 Performance Comparison by Layer on MNLI

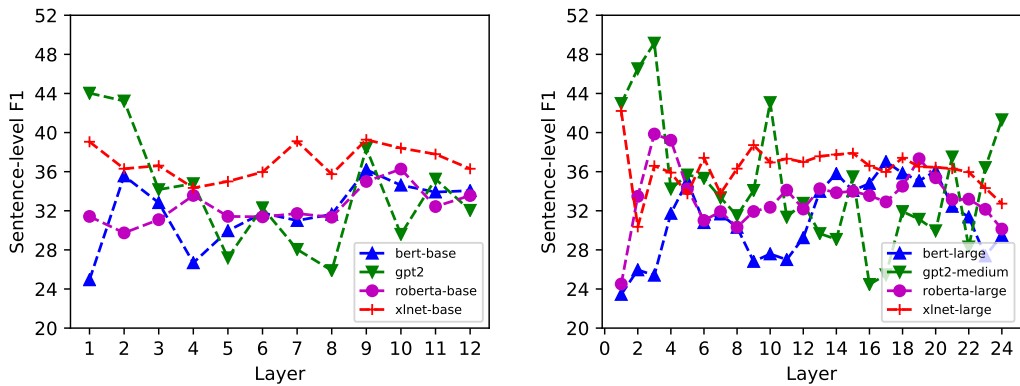

Figure 4: The best layer-wise S-F1 scores of each LM instance on the MNLI test set. (Left) The performance of the X-'base' models. (Right) The performance of the X-'large' models.

## A.5 EXPERIMENTAL RESULTS ON MNLI

Table 4: Results on the MNLI test set. Bold numbers correspond to the top 3 results for each column. L: layer number, A: attention head number (AVG: the average of all attentions). †: Results reported by Htut et al. (2018) and Drozdov et al. (2019). ‡: Approaches in which $\overline{\text{COO}}$ parser is utilized. ∗: These results are not strictly comparable to ours, due to the difference in data preprocessing.

| Model | $f$ | L | A | S-F1 | SBAR | NP | VP | PP | ADJP | ADVP |
|---|---|---|---|---|---|---|---|---|---|---|
| **Baselines** | | | | | | | | | | |
| Random Trees | - | - | - | 21.4 | 11% | 25% | 16% | 22% | 22% | 27% |
| Balanced Trees | - | - | - | 20.0 | 8% | 29% | 11% | 20% | 22% | 32% |
| Left Branching Trees | - | - | - | 8.4 | 6% | 13% | 1% | 4% | 1% | 8% |
| Right Branching Trees | - | - | - | 51.9 | **65%** | 28% | **75%** | 47% | 45% | 30% |
| Random XLNet-base ($F^v$) | - | - | - | 22.0 | 12% | 26% | 15% | 22% | 22% | 25% |
| Random XLNet-base ($F^d$) | - | - | - | 23.5 | 14% | 26% | 18% | 22% | 22% | 25% |
| **Pre-trained LMs (w/o bias)** | | | | | | | | | | |
| BERT-base | HEL | 9 | 10 | 36.1 | 36% | 37% | 34% | 45% | 26% | 42% |
| BERT-large | JSD | 17 | 10 | 37.0 | 38% | 32% | 34% | 50% | 22% | 39% |
| GPT2 | JSD | 1 | 10 | 44.0 | 43% | **53%** | 31% | 60% | 24% | 40% |
| GPT2-medium | JSD | 3 | 12 | 49.1 | 57% | 32% | 61% | 44% | 35% | 37% |
| RoBERTa-base | JSD | 10 | 9 | 36.2 | 26% | 35% | 34% | 50% | 23% | 44% |
| RoBERTa-large | JSD | 3 | 6 | 39.8 | 20% | 28% | 35% | 30% | 28% | 27% |
| XLNet-base | HEL | 1 | 6 | 39.0 | 25% | 39% | 28% | 59% | 35% | 44% |
| XLNet-large | HEL | 1 | 15 | 42.2 | 32% | 49% | 27% | **62%** | 32% | **49%** |
| **Pre-trained LMs (w/ bias $\lambda$=1.5)** | | | | | | | | | | |
| BERT-base | HEL | 2 | 12 | 52.7 | **64%** | 35% | 70% | 50% | **46%** | 30% |
| BERT-large | HEL | 4 | 4 | 51.7 | 63% | 31% | **71%** | 49% | **46%** | 30% |
| GPT2 | HEL | 1 | 10 | 52.2 | 57% | **53%** | 49% | **62%** | 32% | 42% |
| GPT2-medium | HEL | 2 | 1 | **53.9** | 53% | **57%** | 50% | **62%** | 29% | 44% |
| RoBERTa-base | HEL | 2 | 3 | 52.0 | **64%** | 31% | **72%** | 49% | **47%** | 30% |
| RoBERTa-large | L1 | 23 | - | 52.7 | 55% | 40% | 65% | 53% | 43% | 41% |
| XLNet-base | L2 | 8 | - | **54.9** | 57% | 49% | 61% | 55% | 44% | **57%** |
| XLNet-large | L2 | 12 | - | **53.5** | 54% | 47% | 59% | 51% | **48%** | 60% |
| **Other models** | | | | | | | | | | |
| PRPN-UP†‡ | - | - | - | 48.6∗ | - | - | - | - | - | - |
| PRPN-LM†‡ | - | - | - | 50.4∗ | - | - | - | - | - | - |
| DIORA† | - | - | - | 51.2∗ | - | - | - | - | - | - |
| DIORA(+PP)† | - | - | - | 59.0∗ | - | - | - | - | - | - |

## A.6 EXPERIMENTAL DETAILS FOR TRAINING IDEAL DISTANCE MEASURE FUNCTION

In this part, we present the detailed specifications of the experiments introduced in Section 6.2. We assume $f_{\text{ideal}}$ is only compatible with the functions in $G^v$, as the functions in $G^d$ are not suitable for training as the sizes of the representations provided by $G^d$ are variable according to the length of an input sentence. To train the pseudo-optimal function $f_{\text{ideal}}$, we minimize a pair-wise learning-to-rank loss following previous work (Burges et al., 2005; Shen et al., 2018a):

$$L_{\text{dist}}^{\text{rank}} = \sum_{i,j>i} [1 - \text{sign}(d_i^{\text{gold}} - d_j^{\text{gold}})(d_i^{\text{pred}} - d_j^{\text{pred}})]^+, \qquad (3)$$

where $d^{\text{gold}}$ and $d^{\text{pred}}$ are computed from the gold tree and our predicted one, respectively. $[x]^+$ is defined as $max(0, x)$. We train the $f_{\text{ideal}}$ with the PTB training set for 5 epochs. Each batch of the training set contains 16 sentences. We use an ADAM optimizer (Kingma & Ba, 2014) with the learning rate 5e-4. We train the variations of $f_{\text{ideal}}$ differentiated by the choice of $g$ in $G^v$ and report the best result in the Table 2. Each $f_{\text{ideal}}$ is chosen based on its performance on the PTB validation set. Considering the randomness of training, every result for $f_{\text{ideal}}$ is averaged over 3 different trials.

## A.7 CONSTITUENCY TREE EXAMPLES

We randomly select six sentences from PTB and visualize their trees, where the resulting group of trees for each sentence consists of a gold constituency tree and two induced trees (one *without* the right-skewness bias and the other *with* the bias) from our best model—XLNet-base. The 'T' character in the induced trees indicates a dummy tag.

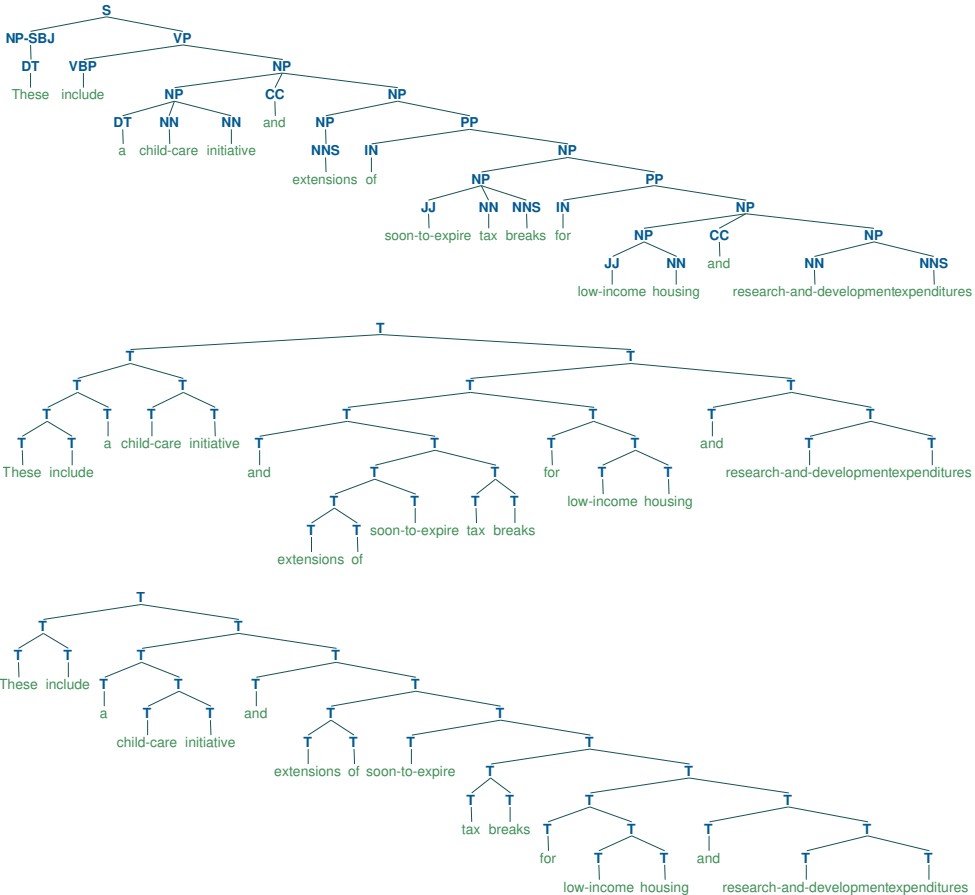

Figure 5: Gold (**top**) and predicted trees (one *without* the bias in the **middle**, the other with the bias at the **bottom**) for the sentence 'These include a child-care initiative and extensions of soon-to-expire tax breaks for low-income housing and research-and-development expenditures'.

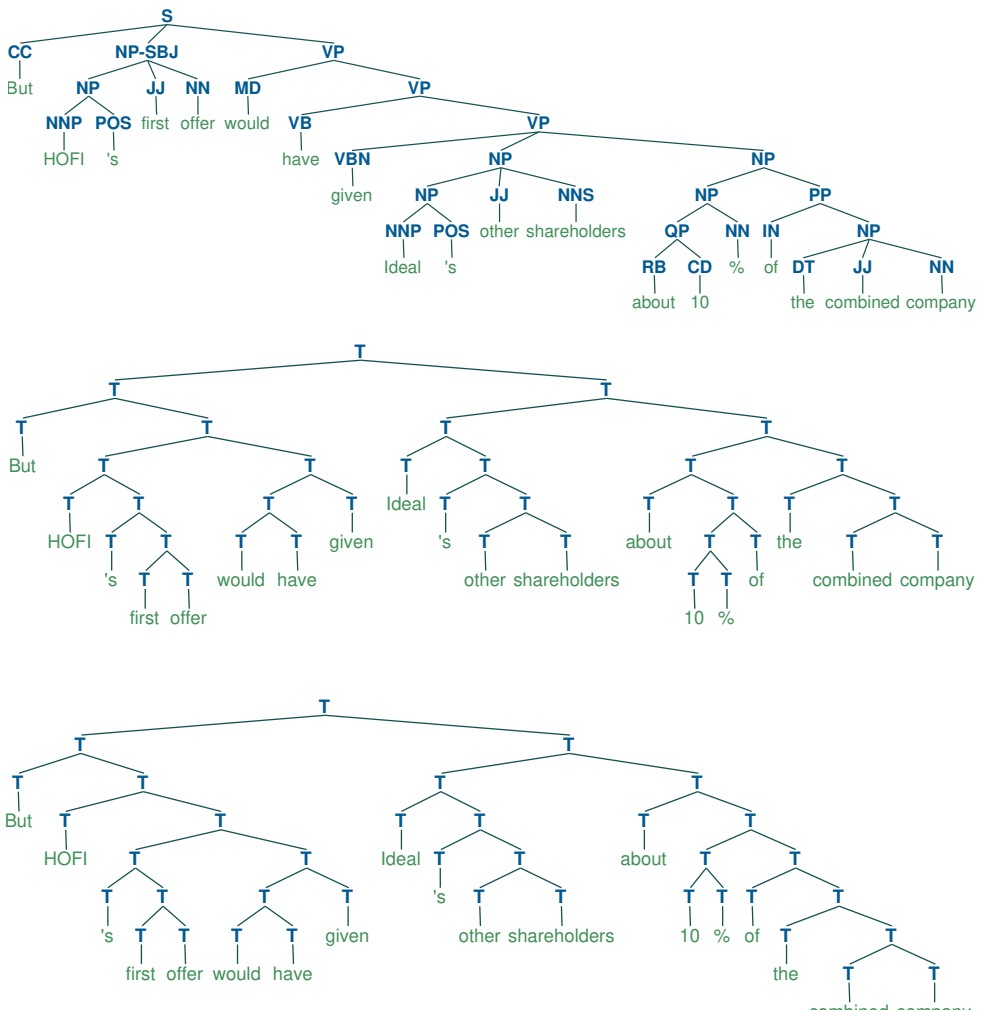

Figure 6: Gold (**top**) and predicted trees (one *without* the bias in the **middle**, the other with the bias at the **bottom**) for the sentence 'But HOFI 's first offer would have given Ideal 's other shareholders about 10 % of the combined company'.

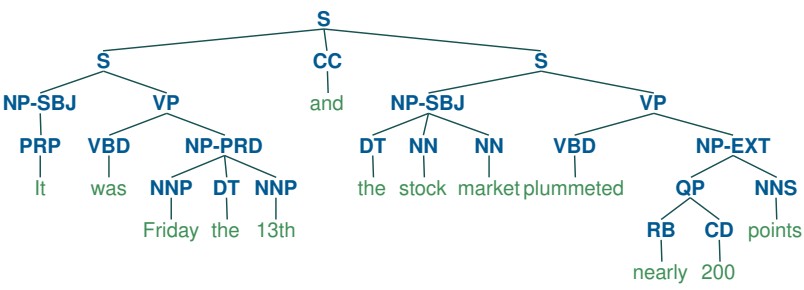

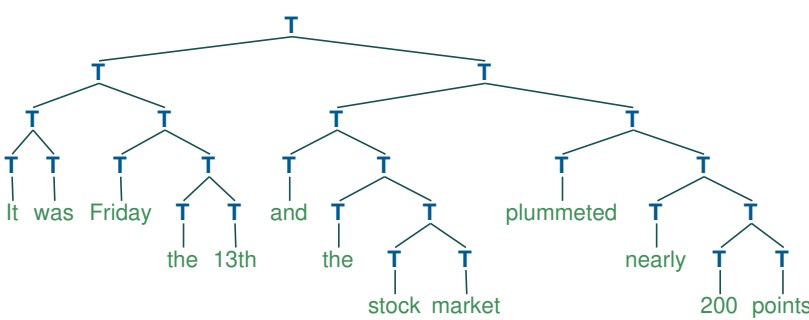

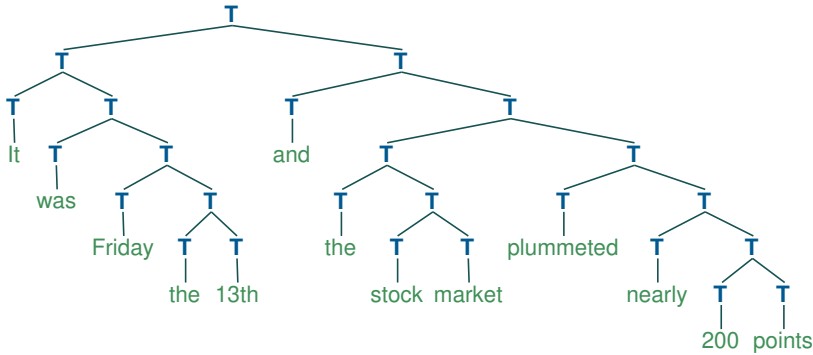

Figure 7: Gold (**top**) and predicted trees (one *without* the bias in the **middle**, the other with the bias at the **bottom**) for the sentence 'It was Friday the 13th and the stock market plummeted nearly 200 points'.

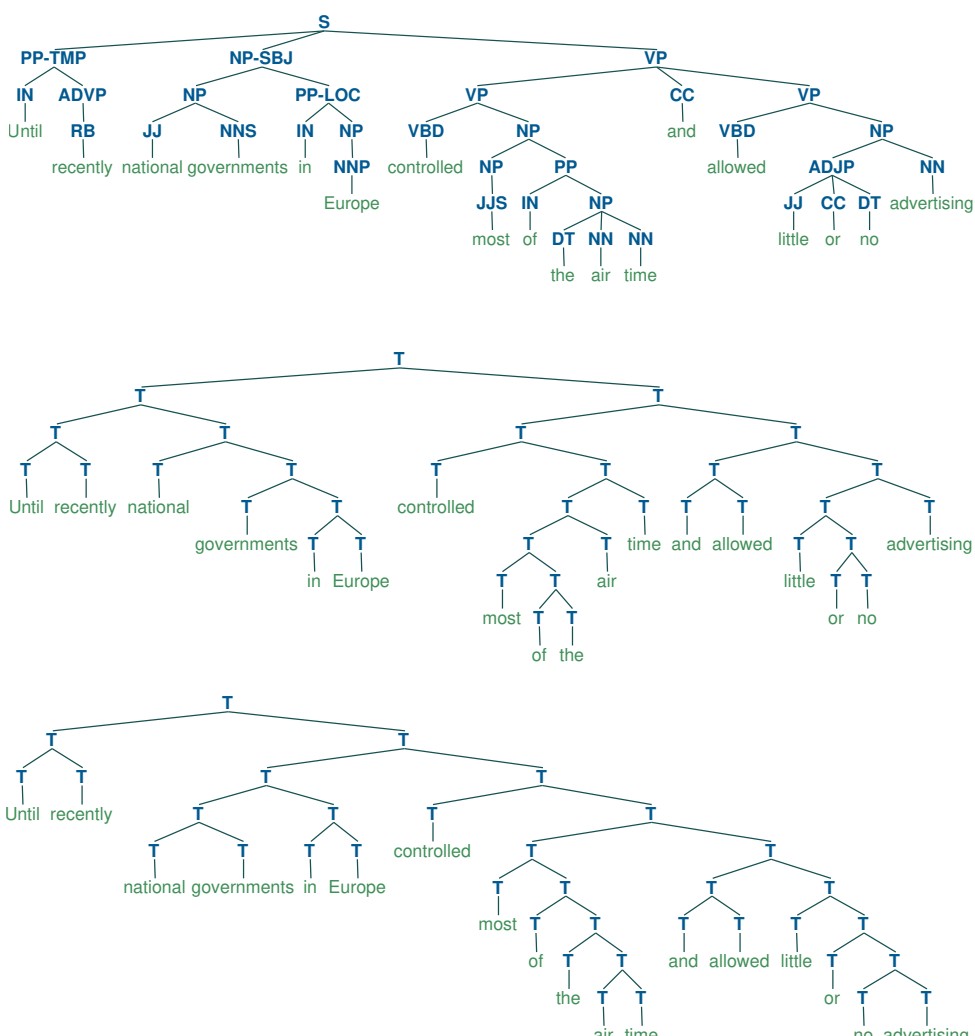

Figure 8: Gold (**top**) and predicted trees (one *without* the bias in the **middle**, the other with the bias at the **bottom**) for the sentence 'Until recently national governments in Europe controlled most of the air time and allowed little or no advertising'.

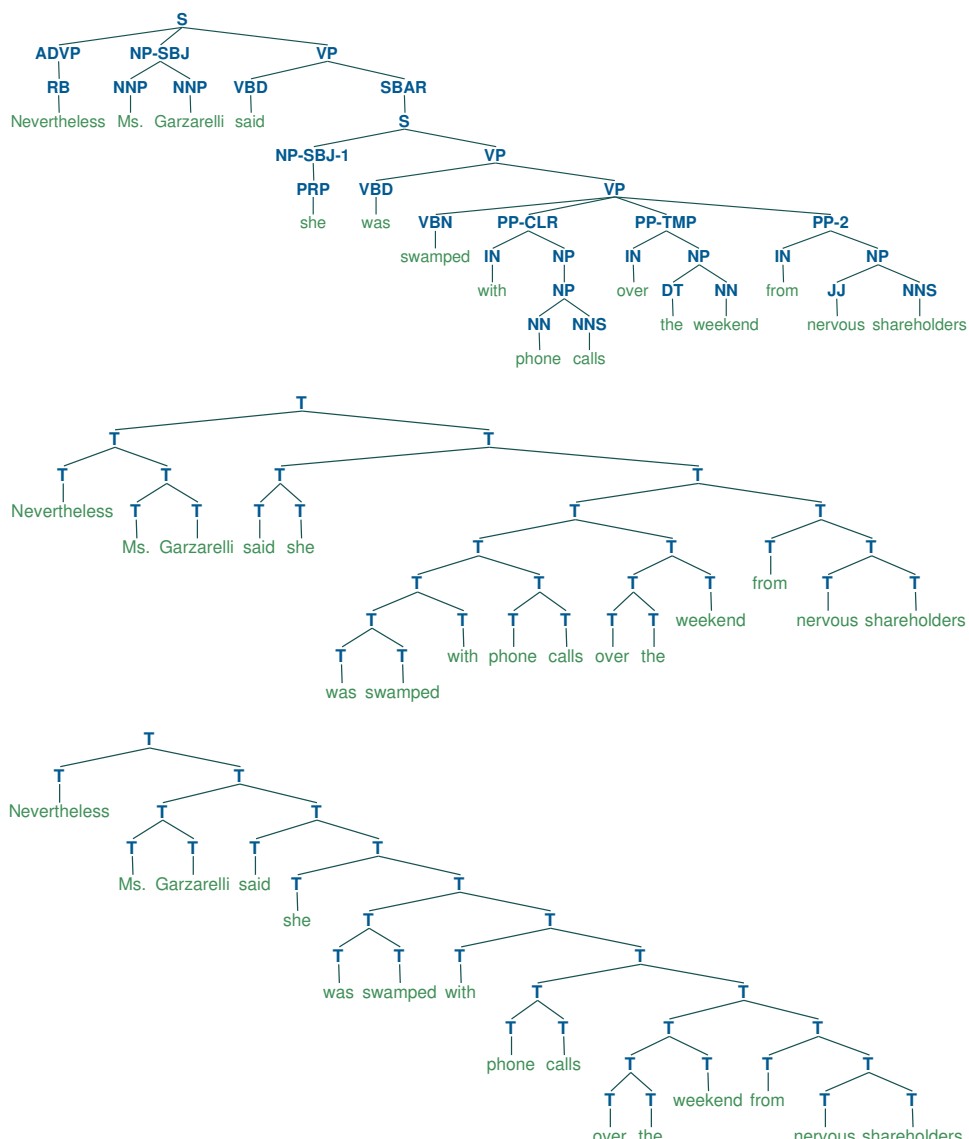

Figure 9: Gold (**top**) and predicted trees (one *without* the bias in the **middle**, the other with the bias at the **bottom**) for the sentence 'Nevertheless Ms. Garzarelli said she was swamped with phone calls over the weekend from nervous shareholders'.

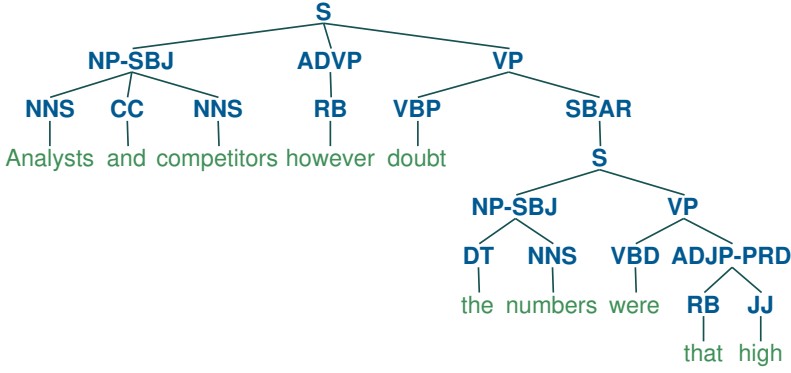

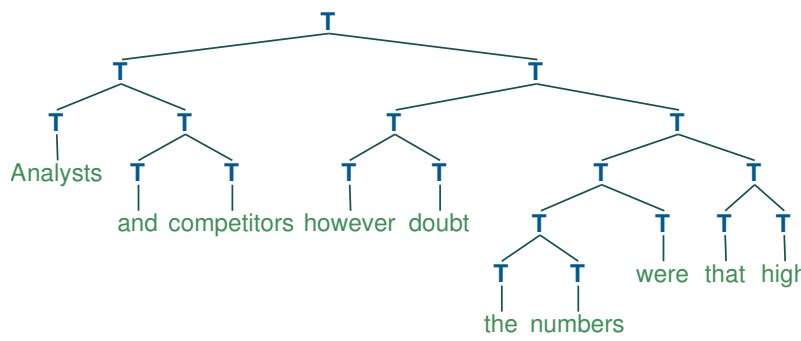

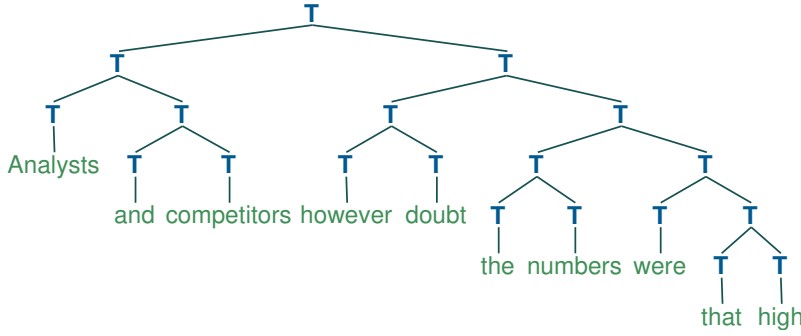

Figure 10: Gold (**top**) and predicted trees (one *without* the bias in the **middle**, the other with the bias at the **bottom**) for the sentence 'Analysts and competitors however doubt the numbers were that high'.

