# OpenReview forum: "Are Pre-trained Language Models Aware of Phrases? Simple but Strong Baselines for Grammar Induction"
_ICLR.cc/2020/Conference — Accept (Poster)_

### Official Review · AnonReviewer3 · 2019-10-06
**Official Blind Review #3**

**Rating:** 8

**Review:**

[EDIT: Thank you very much for the thoughtful and extensive response. Based on the latest iteration of the paper, I am changing my score to 8]

This paper studies the representations learned by large pre-trained models trained on language modeling objective (or language modeling-like objective, in the case of masked models). In particular, the authors investigate whether constituency information is present in the hidden layers. In contrast to much existing work that probe for this information with (for example) linear models on top of hidden representations, the authors propose to directly extract binary trees from the model using "syntactic-distance" measure that is calculated in various ways (e.g. dot product of hidden representations, distance in attention distributions). Across various models, the authors find that it is indeed possible to induce linguistically meaningful trees, in particular outperforming a right-branching baseline that is strong for English

I found this to be a creative alternative to the existing "BERTology" type papers that rely mostly on linear probes, and the experiments are done across a wide number of setups (e.g. across models/datasets).  However, I have several questions/issues with the paper if addressed, would make the paper much stronger:

- The degrees of freedom afforded by the choice in number of layers, heads, similarity measure, etc. is quite high. For example, with 24 layers and 16 attention heads I count (24*(16+1)*2 + 24*3) = 888 different measures of distance that could be used to induce trees. I think the authors should make sure that they are not "overfitting" to the test set with the following set of tests:

1. Only use the training set to select the layer/metric combination.
2. See if the selected combination generalizes to other languages (from comparing Table 1 against Table 2, it seems like this doesn't even generalized within the same language across different domains? This is worrying).
3. See what the performance is with random initialization. In particular, I would like to see two setups: (a) the network is randomly initialized, (b) the *representations* (i.e. attention/hidden states) are randomly initialized. The random initialization should have variance such that it's not just uniform distributions.

- I found the right branching bias baseline experiment not very informative. As the authors allude to, performance on unsupervised parsing itself is not so interesting, and the point of the paper (in my view) is not to get the best unsupervised parsing performance.

- Related to the above, it is known that the decoding algorithm of Shen et al. 2018 is itself heavily biased towards right branching trees. See in particular: https://arxiv.org/pdf/1909.09428.pdf. Given this, I am not sure if the supposedly good performance in parsing is due to the model actually learning constituency or just the bias.

- It would have been interesting to see how the performance changes as:
1. The model is fine-tuned on the PTB-training set.
2. The model is trained from scratch on PTB (obviously with a much smaller model since PTB is smaller).




**Experience Assessment:**

I have published one or two papers in this area.

**Review Assessment: Checking Correctness Of Derivations And Theory:**

I assessed the sensibility of the derivations and theory.

**Review Assessment: Checking Correctness Of Experiments:**

I assessed the sensibility of the experiments.

**Review Assessment: Thoroughness In Paper Reading:**

I read the paper at least twice and used my best judgement in assessing the paper.

---

> ### Author Response · Authors · 2019-11-13
> **Response to the comments from Reviewer 3: (1)**
>
> Thanks for dedicating your time for reviewing our paper and providing thoughtful comments.
>
> We have uploaded our revision to reflect your (and other reviewers') opinions.
> It would be appreciated if you read the revision before reading our responses, which are presented below.
>
> Q1: Only use the training set to select the layer/metric combination.
>
> A: Thanks to the comments from reviewers, we have realized that there were some mistakes and ambiguity in our experimental settings.
> We here share the newly defined experimental settings used in our revision. (The details are specified in Section 5.1.)
>
> 1. We decide **not** to employ the trick which exists in the source codes for Shen et al. 2018 (PRPN) encouraging the resulting constituency trees to be more right-skewed because we can directly adjust the bias using our method proposed in Section 4.4 if needed. (Footnote 1).
> 2. For each LM, we tune the best combination of "f" and "g" functions using the validation set. Then, we derive a set of syntactic distances "d" for sentences in the test set using the chosen functions, followed by the resulting constituency trees converted from each distance by the tree construction algorithm. We also use the validation set to tune the lambda (bias hyperparameter).
> 3. We report new experimental results in our revision. If there exist changes in numbers, they are colored in red. We notice that there are no major changes in numbers and in the best selection of f and g, even though some experimental protocols are modified.
>
> Q2: See if the selected combination generalizes to other languages (from comparing Table 1 against Table 2, it seems like this doesn't even generalized within the same language across different domains? This is worrying).
>
> A: We understand your concern as generalization is definitely one of the big issues when developing data-driven ML methods.
> There are actually many candidates we can consider in our framework even though we mostly report the best cases, and it is hard to imagine that a specific selection for "f" and "g" is always superior to other choices across different domains.
> However, we empirically observed that the best combinations of f and g in PTB also record comparable scores in MNLI (even though not the best), implying that there are some specific combinations of "f" and "g" which are broadly sensitive to syntactic information irrespective of domains.
> Furthermore, we identify from our experimental results that the best selection of "f" and "g" for the XLNet-base model is always identical across domains.
> Considering the above facts, it seems persuasive to say that our approach is robust to some degree at least in limited conditions (when relying on the XLNet, etc.).
>
> However, one limitation of our work, as we already mentioned in the paper, is that we have only experimented on English datasets. Therefore, we think expanding this work to other languages should be regarded as one of the top priority topics to be dealt with in our future work.
>
> Q3. See what the performance is with random initialization. In particular, I would like to see two setups: (a) the network is randomly initialized, (b) the *representations* (i.e. attention/hidden states) are randomly initialized. The random initialization should have variance such that it's not just uniform distributions.
>
> A: Following your advice, we introduce two more baselines (in Section 5.1.2) which are identical to our models except that their "g" functions are based on randomly initialized Transformer---one having the same configuration as the XLNet-base model---rather than pre-trained ones.
> To be concrete, we provide a baseline named `Random Transformer ("F^v")' which applies the functions in "F^v" on random hidden representations and `Random Transformer ("F^d")' that utilizes the functions in "F^d" and random attention distributions, respectively.
> Considering the randomness of initialization and possible choices for "f", the final score for each of the baselines is calculated as an average over 5 trials of each possible "f", i.e., an average over 5 * 3 (L1, L2, COS) = 15 runs in case of "F^v" and 5 * 2 (HEL, JSD) = 10 runs for "F^d".
> These baselines enable us to estimate the exact advantage we obtain by pre-training LMs, effectively removing additional unexpected gains including one inherited from the intrinsic bias of the parsing algorithm.

---

> ### Author Response · Authors · 2019-11-13
> **Response to the comments from Reviewer 3: (2)**
>
> Q4: I found the right branching bias baseline experiment not very informative. As the authors allude to, performance on unsupervised parsing itself is not so interesting, and the point of the paper (in my view) is not to get the best unsupervised parsing performance. Related to the above, it is known that the decoding algorithm of Shen et al. 2018 is itself heavily biased towards right branching trees. See in particular: https://arxiv.org/pdf/1909.09428.pdf. Given this, I am not sure if the supposedly good performance in parsing is due to the model actually learning constituency or just the bias.
>
> A: In our revision, we have rewritten Section 4.4 to clearly define the purpose and necessity of introducing the bias.
> The main purpose of introducing such a bias is examining what changes are made to the resulting tree structures rather than boosting quantitative performances per se, though it is of note that it serves this purpose as well.
> We believe that this additional consideration is necessary based on two points.
> First, English is what is known as a head-initial language.
> That is, given a selector and argument, the selector has a strong tendency to appear on the left, e.g. `eat food', or `to Canada'.
> By adjusting the bias injected into syntactic distances derived from pre-trained LMs, we can figure out whether the LMs are capable of inducing the right-branching bias, which is one of the main properties of English syntax; if injecting the bias does not influence the performance of the LMs on unsupervised parsing, we can conjecture they are inherently capturing the bias to some extent.
> Second, as mentioned above, we have witnessed some previous work (PRPN, ON) where the right-skewness bias is exploited, although it could be regarded as not ideal.
> What we intend to focus on is the question about which benefits the bias provides for such parsing models, leading to overall performance improvements.
> In other words, we look for what the exact contribution of the bias is when it is injected into grammar induction models, by explicitly controlling the bias using our framework.
>
> Moreover, in Section 5.2, we report that by explicitly controlling the bias through our framework and observing the performance gap between our models with and without the bias, we confirm that the main contribution of the bias comes from its capability to capture subordinate clauses (SBAR) and verb phrases (VP).
> This observation provides a hint for what some previous work on unsupervised parsing desired to obtain by introducing the bias to their models.
> It is intriguing to note that all of the existing grammar induction models are inferior to the right-branching baseline in recognizing SBAR and VP (although some of them already implicitly utilized the right-skewness bias), implying that the same problem---models do not properly capture the right-branching nature of English syntax---may also exist in current grammar induction models.
>
> In addition, we claim that we are well-aware of the concerns in the community about unreasonable gains obtained by exploiting the bias in unsupervised parsing, and that's why we decide to utilize the bias in our work paradoxically, aiming to better understand the exact points where the bias provides benefits.
> Regarding your concern on the incompleteness of the parsing algorithm we use (recently reported by Dyer et al., 2019, as you mentioned), we added a footnote (Footnote 6) in the revision to explain that our experimental setting (i.e., comparing with Random Transformers) is a good countermeasure to exclude the influence of the parsing algorithm.
>
> Lastly, we also agree with your opinion that using trees extracted from the biased models on further analysis (Section 6) can mislead the final results.
> To this end, we decided not to use the models with bias in Section 6.1, and provide additional tree examples derived from our model without the bias, facilitating visual comparisons between trees with and without the bias.
>
> Q5: It would have been interesting to see how the performance changes as:
> 1. The model is fine-tuned on the PTB-training set.
> 2. The model is trained from scratch on PTB (obviously with a much smaller model since PTB is smaller).
>
> A: It's actually a good point as fine-tuning on the PTB domain would lead models to adapt to the said domain and show better performance.
> However, our main contribution in this work is to discover the universal capability of language models obtained by pre-training the models on large corpora, and it is not our main concern to fine-tune the LMs.
> But, we agree with your opinion that the settings you proposed would be very interesting to investigate and we leave this as future work.
>
> We hope our answers would be clear enough to resolve your questions/concerns.
> Please let us know whenever you have any other questions/comments or things to discuss.
>
> Thanks again for your kind and thoughtful advice!

---

### Official Review · AnonReviewer2 · 2019-10-23
**Official Blind Review #2**

**Rating:** 6

**Review:**


*Summary

This paper describes an effective method that induces constituency trees from pre-trained language models, which are attracting great attention recently. The authors demonstrated that the pre-trained language models have some properties that are similar to constituency grammar by showing some interesting features of the extracted trees.
This study is based on the motivation to unveil the reason why such pre-trained language models work and the extent to which pre-trained language models capture the syntactic notion of the constituency.
They also show that the method can become a reasonable baseline method for English grammar induction.

*Decision and supporting arguments

I'm leaning toward accepting the paper as a conference paper.
As the author describes, pre-training LMs are attracting attention, and many people want to understand their inner workings. The paper provides a systematic analysis of the pre-trained LMs from the viewpoint of grammar induction. Also, this paper is well-structured and offers a good literature review.  I think the paper has enough value to be published from the scientific viewpoint.


*Additional feedback

Figure 1 & 5-10 look rasterized. It's better to use vector images.


**Experience Assessment:**

I have read many papers in this area.

**Review Assessment: Checking Correctness Of Derivations And Theory:**

N/A

**Review Assessment: Checking Correctness Of Experiments:**

I assessed the sensibility of the experiments.

**Review Assessment: Thoroughness In Paper Reading:**

I made a quick assessment of this paper.

---

> ### Author Response · Authors · 2019-11-13
> **Response to the comments from Reviewer 2:**
>
>
> Thanks for dedicating your time for reviewing our paper and providing comments.
>
> We have revised our paper so that every picture in the paper is not rasterized (we use pdf or svg formats).
> Please check out our revision and let us know if you have any other comments on our paper.
> Briefly speaking, we have modified our paper to reflect helpful comments from other reviewers, for example, clarifying experimental protocols, providing some more baselines on experiments to reveal the exact contribution of pre-training, and adding detailed comments about our experimental results.
>
> Thanks again for your kind and thoughtful advice!

---

### Official Review · AnonReviewer1 · 2019-10-23
**Official Blind Review #1**

**Rating:** 6

**Review:**

I am satisfied with the author's response and I am increasing the score to 6 after the rebuttal.

===

This paper introduces a new and simple method of probing whether syntax information under the form of constituency trees is present in recent pre-trained language models (e.g. BERT, RoBERTa, XLNet and GPT2) without any additional task-specific training. They use the previously proposed concept of "syntactic distance" [1] as a sufficient statistics for the constituency tree of a given sentence. They compute the distance between neighboring words in sentence using a distance function f(g(w_i), g(w_i+1)) where f is a divergence between distributions and g(w_i) is the self-attention distribution of w_i at a given layer: the intuition is that words that have similar self-attention distributions belong to the same constituent and thus their syntactic distance is small. In order to conform with right-skewness of English syntax, they propose an affine transformation of the distances that encourage right-skewed trees, and tune the hyper-parameter directly on the target F1 score wrt ground-truth trees. Results suggest that large, pretrained models capture constituency trees to some extent.

This is an empirical and analysis paper which can be considered as the “twin” of Hewitt et. al (2019, analyzes whether dependency structure can be probed from large pre-trained models). The paper is well-written and easy to understand. The experiments are in general well-organized even if they lack clarity at some point. The related work is comprehensive and covers most of the work in the domain of grammar induction and unsupervised parsing. In its current form however, I feel like the paper misses a "second half" therefore it isn’t quite above the acceptance bar. I'd be happy if the authors could kindly elaborate on the following major weaknesses: (i) unclear scientific motivation of using the right-skewness bias; (ii) lack of clarity in the experimental results; (iii) lack of depth and perspective.

1) About right-skewness bias:
I appreciate that the authors clearly guard against using explicit right-skewness biases. However, I am not sure why the authors use of Eq. (2) apart from boosting performance. The authors claim that "the main purpose of explicitly injecting such a bias is examining what changes are made to the resulting tree structures" but the purposed changes are not discussed at length in the text. Therefore:
1.1) What's the scientific motivation (apart from boosting performance) of skewing the trees using Eq. (2) ?
1.2) What can you infer from trees without bias and tree with bias ?

2) About clarity in the experimental results:
2.1) (Shen, 2018) use an "implicit" right-skewness bias. From the text, it seems that your results w/o bias use the same bias as (Shen, 2018) ? Therefore they actually have a bias ? This is a bit confusing. What about using the unbiased parsing algorithm as presented in "Straight to the tree..." ?
2.2) This sentence is unclear: "we select one derived from the best choice of f and g in terms of sentence-level F1 (S-F1)
w.r.t. gold-standard trees as a representative for the LM". Are you using S-F1 on *training* for tuning f and g ?
2.3) This sentence is unclear: "As LMs are not fine-tuned with training sets in our framework, we only use the test
set of the PTB", what do you mean by "using the test set" ? Do you mean evaluate or use it for tuning (cf 2.2) ?

3) About perspective and depth:
3.1) The fact that ADVP , ADJP performance is higher for the tested models is interesting. Do you have any hypothesis for why this is happening?
3.2) Right-skewness seems to mainly help with VP. Can you formulate an hypothesis for why current models underperform on this label or do not show right bias?
3.3) The results on Table 2 are interesting, but imho, they are basically showing that these models cannot do constituency parsing with just a linear probe on top of the representations and using the syntactic distance algorithm?
3.4) Did you consider using other parsing algorithms e.g. chart parser, instead of just the syntactic distance ? That would make a stronger paper.

**Experience Assessment:**

I have published one or two papers in this area.

**Review Assessment: Checking Correctness Of Derivations And Theory:**

I assessed the sensibility of the derivations and theory.

**Review Assessment: Checking Correctness Of Experiments:**

I carefully checked the experiments.

**Review Assessment: Thoroughness In Paper Reading:**

I read the paper at least twice and used my best judgement in assessing the paper.

---

> ### Author Response · Authors · 2019-11-13
> **Response to the comments from Reviewer 1: (1)**
>
>
> Thanks for dedicating your time for reviewing our paper and providing thoughtful comments.
>
> We have uploaded our revision to reflect your (and other reviewers') opinions.
> It would be appreciated if you read the revision before reading our responses, which are presented below.
>
> 1.1) Q: "What's the scientific motivation (apart from boosting performance) of skewing the trees using Eq. (2)?
>
> A: We added a paragraph in Section 4.4 that clearly explains the reason why we should consider injecting the bias.
> Basically, there are two perspectives that explain its necessity.
> First, English is what is known as a head-initial language.
> That is, given a selector and argument, the selector has a strong tendency to appear on the left, e.g. `eat food', or `to Canada'.
> Head-initial languages, therefore, have an in-built preference for right-branching structures.
> By adjusting the bias injected into syntactic distances derived from pre-trained LMs, we can figure out whether the LMs are capable of inducing the right-branching bias, which is one of the main properties of English syntax; if injecting the bias does not influence the performance of the LMs on unsupervised parsing, we can conjecture they are inherently capturing the bias to some extent.
> Second, we have witnessed some previous work where the right-skewness bias is exploited, even though it could be regarded as not ideal.
> What we intend to focus on here is the question about which benefits the bias can provide for such parsing models leading to overall performance improvements.
> In other words, we look for what the exact contribution of the right-skewness bias is when it is injected into grammar induction models, by explicitly controlling the bias using our framework.
>
> 1.2) Q: What can you infer from trees without bias and tree with bias ?
>
> A: As specified in Section 5.2 of our revision, by explicitly controlling the bias through our framework and observing the performance gap between our models with and without the bias, we find that the main contribution of the bias comes from its capability to capture subordinate clauses (SBAR) and verb phrases (VP). Moreover, in Section 6.3, we additionally provide tree examples derived from our model without the bias, facilitating visual comparisons between trees with and without the bias.

---

> ### Author Response · Authors · 2019-11-13
> **Response to the comments from Reviewer 1: (2)**
>
>
> 2.1) Q: (Shen, 2018) use an "implicit" right-skewness bias. From the text, it seems that your results w/o bias use the same bias as (Shen, 2018) ? Therefore they actually have a bias ? This is a bit confusing. What about using the unbiased parsing algorithm as presented in "Straight to the tree..." ?
> 2.2) Q: This sentence is unclear: "we select one derived from the best choice of f and g in terms of sentence-level F1 (S-F1)
> w.r.t. gold-standard trees as a representative for the LM". Are you using S-F1 on *training* for tuning f and g ?
> 2.3) Q: This sentence is unclear: "As LMs are not fine-tuned with training sets in our framework, we only use the test
> set of the PTB", what do you mean by "using the test set" ? Do you mean evaluate or use it for tuning (cf 2.2) ?
>
> A: Thanks to the comments from reviewers, we recognize that there were indeed some mistakes and ambiguity in our experimental settings.
> We here share the newly defined experimental settings used in our revision. (The details are specified in Section 5.1.)
>
> 1. We decide **not** to employ the trick which exists in the source codes for Shen et al. 2018 (PRPN) encouraging the resulting constituency trees to be more right-skewed because we can directly adjust the bias using our method proposed in Section 4.4 if needed. (Footnote 1).
> 2. For each LM, we tune the best combination of "f" and "g" functions using the validation set. Then, we derive a set of syntactic distances "d" for sentences in the test set using the chosen functions, followed by the resulting constituency trees converted from each distance by the tree construction algorithm. We also use the validation set to tune the lambda (bias hyperparameter).
> 3. We report new experimental results in our revision. If there exist changes in numbers, they are colored in red. We notice that there are no major changes in numbers and in the best selection of f and g, even though some experimental protocols are modified.
>
> 3.1) Q: The fact that ADVP , ADJP performance is higher for the tested models is interesting. Do you have any hypothesis for why this is happening?
>
> A: We added detailed descriptions of our own observation and hypothesis about the phenomenon in the revision (in Section 5.2) to provide deeper intuition.
> Specifically, to further explain why some pre-trained LMs are good at capturing ADJPs and ADVPs, we manually investigated the attention heatmaps of the sentences that contain ADJPs or ADVPs.
> From the inspection, we empirically found that there are some keywords---including `two', `ago', `too', and `far'---that have different patterns of attention distributions compared to those of their neighbors and that these keywords can be a clue for our framework to recognize the existence of ADJPs or ADJPs.
> It is also worth mentioning that ADJPs and ADVPs consist of a relatively smaller number of words than those of SBAR and VP, indicating that the LMs combined with our method have strength in correctly finding small chunks of words, i.e., low-level phrases.
>
> 3.2) Q: Right-skewness seems to mainly help with VP. Can you formulate an hypothesis for why current models underperform on this label or do not show right bias?
>
> A: We also additionally mentioned our conjecture about this phenomenon in our revision (in Section 5.2).
> One possible assumption is that models do not need the bias to perform well in language modeling, although future work should provide a rigorous analysis that shows why existing grammar induction models seem to not mirror the right-branching bias.
>
> 3.3) Q: The results on Table 2 are interesting, but imho, they are basically showing that these models cannot do constituency parsing with just a linear probe on top of the representations and using the syntactic distance algorithm?
>
> A: We want to emphasize that the results with a pseudo-optimum "f" in Table 2 came from training with **gold-standard trees** from PTB.
> As we mainly focus on the unsupervised setting, where we presume that such supervision is unfeasible, we are not able to easily get "f_ideal" in principle.
> There would be, of course, a more sophisticated "f" than the simple ones we introduced, but it requires dedicated training procedures and complex algorithms to derive, which we intend to evade in this work.
> Furthermore, we note that our method, which employs a relatively simple "f", is competitive enough to be utilized as a baseline, showing decent performance in all evaluation metrics when compared against a linear model trained **with supervision from the gold-standard trees**.

---

> ### Author Response · Authors · 2019-11-13
> **Response to the comments from Reviewer 1: (3)**
>
>
> 3.4) Q: Did you consider using other parsing algorithms e.g. chart parser, instead of just the syntactic distance ? That would make a stronger paper.
>
> A: Yes, that’s actually a good point, because a different parsing algorithm including chart parsing could result in a different outcome.
> However, in this work, we would like to focus on extracted trees from syntactic distances computed from pre-trained LMs, and we leave utilizing other parsing algorithms as future work.
> Here are two main reasons for our decision.
> First, leveraging the concept of syntactic distance enables us to compare our models against the existing grammar induction models more directly, as a considerable amount of previous work, such as PRPN and ON, actually relies on this concept.
> Second, as one of our main purposes in this paper is to propose a way of deriving a partly reliable phrase-structure tree on the fly (without training), we are reluctant to utilize complex algorithms that may require heavy computations and complex training procedures.
>
> We hope our answers would be clear enough to resolve your questions/concerns.
> Please let us know whenever you have any other questions/comments or things to discuss.
>
> Thanks again for your kind and thoughtful advice!

---

### Decision · Program_Chairs · 2019-12-19

**Decision:**

Accept (Poster)

**Comment:**

This paper presents results of looking at the inside of pre-trained language models to capture and extract syntactic constituency. Reviewers initially had neutral to positive comments, and after the author rebuttal which addressed some of the major questions and concerns, their scores were raised to reflect their satisfaction with the response and the revised paper. Reviewer discussions followed in which they again expressed that they became more positive that the paper makes novel and interesting contributions.

I thank the authors for submitting this paper to ICLR and look forward to seeing it at the conference..